# The Removal of Strontium Ions from an Aqueous Solution Using Na-A Zeolites Synthesized from Kaolin

**DOI:** 10.3390/ma17030575

**Published:** 2024-01-25

**Authors:** Woo-Ri Lim, Chang-Han Lee, Chung-Mo Lee

**Affiliations:** 1KNU LAMP Research Center, KNU Institute of Basic Sciences, College of Natural Sciences, Kyungpook National University, Daegu 41566, Republic of Korea; wooriful@naver.com; 2Department of Environmental Administration, Catholic University of Pusan, Busan 46252, Republic of Korea; 3Department of Geological Sciences and Institute for Future Earth, Pusan National University, Busan 46241, Republic of Korea

**Keywords:** adsorption, Na-A zeolite, crystallinity, radioactive waste, strontium

## Abstract

Sr^2+^ ions in an aqueous solution were removed using Na-A zeolites synthesized from kaolin, a natural mineral. Na-A zeolites with high crystallinity were synthesized using NaOH/kaolin mass ratios of 0.6 (ZK06) and 0.9 (ZK09). The adsorption reached equilibrium within 120 min. The adsorption data obtained from experiments for Sr^2+^ using ZK06 and ZK09 were appropriately analyzed with pseudo-second-order kinetic and Langmuir isotherm models. Comparing the maximum adsorption capacities (q_m_) of ZK06 and ZK09 for Sr^2+^, the highest values were obtained at 1.90 and 2.42 mmol/g, respectively. Consequently, the Na-A zeolites synthesized from kaolin can be evaluated as adsorbents with high adsorption capacities for the removal of Sr^2+^, proportional to the degree of their crystallinity.

## 1. Introduction

Nuclear power plants (NPPs) are actively operated worldwide to generate electricity, and their radioactive waste is considered a hazardous contaminant. The accumulation potential of radioactive elements in animals and plants causes serious human health and environmental problems owing to their relatively long half-lives [1,2]. Radioactive waste is generated in a wide range of industries, such as nuclear power plants, mining, and medicine, and large amounts of strontium are present in such waste [3]. Strontium can be accumulated in the human body, which can cause bone cancer and leukemia because it is a highly poisonous radioactive materials [4].

Zeolites are aluminosilicate structures consisting of tetrahedral SiO_4_ and AlO_4_ linked with shared oxygen bridges that form cages that can capture diverse hydrated ions depending on their size [5,6]. The advantages of using zeolites in the safe treatment of radioactive waste are their high selectivity and excellent radiation resistance [7]. Na-A zeolite is one of the low-silica zeolites with a Si/Al content ratio of 1 and a cubic crystal structure [8]. While high-alumina zeolites are unsuitable as catalysts due to their lack of acidity, Na-A zeolites are appropriate for adsorption reactions due to the large density of cation-capturing sites in the framework. Owing to their characteristically high cation-exchange capacity (CEC), Na-A zeolites are widely used for removing toxic metal ions and radionuclides from aqueous solutions in wastewater treatment. In particular, since the pore apertures of Na-A zeolites are in the range of 0.23 nm to 0.42 nm, they can be considered to have a framework structure suitable for capturing Sr^2+^, which has a hydration radius of approximately 0.41 nm [9].

Among the various synthesis methods for producing Na-A zeolites, hydrothermal and fusion/hydrothermal methods are widely used in the synthesis of inexpensive raw materials such as coal fly ash, blast-furnace slag, waste sludge, lithium slag, and natural minerals [10]. Kaolin is a low-cost raw material and is abundant across the world. When kaolin is used as a synthetic raw material, Na-A zeolites can be obtained by dissolving metakaolin in a sodium hydroxide (NaOH) solution without adding any other silica or alumina sources [11].

Researchers have developed approaches to evaluate methods for removing radioactive elements using various zeolites as adsorbents. Smiciklas et al. [12] fitted their experimental results to the Langmuir isotherm model and demonstrated that the adsorption capacities of natural zeolites for Cs^2+^, Co^2+^, and Sr^2+^ were 0.03, 0.56, and 0.11 mmol/g, respectively. El-Kamash used the pseudo-second-order kinetic and Langmuir models and showed that the removal efficiency of synthetic Na-A zeolite for Sr^2+^ was 0.79 mg/g [13]. Lee et al. performed the Sr^2+^ removal in an aqueous solution using synthesized Na-A zeolite from coal fly ash (CFA), and the process presented an adsorption capacity of 1.78 mmol/g [14]. Despite several studies, there is a lack of research on the Sr^2+^ adsorption characteristics of zeolites based on their crystallinity.

In a previous study, it was confirmed that kaolin, an inexpensive and natural clay mineral, was successfully converted into Na-A zeolites using a fusion/hydrothermal method [15]. In this study, we focused on the removal of Sr^2+^ from aqueous solutions and evaluated the adsorption capacities of synthesized Na-A zeolites as adsorbents. The experimental data were analyzed by model fitting with adsorption kinetic and isotherm models to evaluate the characteristics of adsorption for Sr^2+^ on the samples. For adsorption kinetics, the pseudo-first- and second-order models were applied, while the Langmuir and Freundlich models were employed for the adsorption isotherm.

## 2. Materials and Methods

### 2.1. Materials

Na-A zeolites were synthesized from natural kaolin using the fusion/hydrothermal method with different NaOH/kaolin mass ratios of 0.6 and 0.9. These samples were named ZK06 and ZK09, respectively. They had a cubic morphology with mean particle sizes of 1.25–1.79 μm (Figure 1, [15]).

To determine that the crystal structures of ZK06 and ZK09 were formed through the synthesis process, their X-ray diffraction (XRD) patterns were compared with the commercial Na-A zeolite as the standard. Figure 2 shows that as the NaOH/kaolin mass ratio increases, its XRD pattern and peak intensity become closer to those of the commercial Na-A zeolite (H_74_Al_11.8_Na_13.1_O_85_Si_12.2_). When the XRD pattern of ZK06 has a low NaOH/kaolin mass ratio, low-intensity peaks remain, suggesting the presence of amorphous components that have not yet been crystallized.

The crystallinity for evaluating the synthesis efficiency of the sample was approximated by taking the ratio of the sum of the integrated peak areas of the sample to the sum of the integrated peak areas of commercial Na-A zeolite that appear intrinsically at 2θ values of 7.20°, 10.18°, 12.46°, 16.10°, 21.66°, 23.98°, 26.10°, 27.10°, 29.92°, 30.82°, 32.52°, and 34.16° [16,17]. The degree of crystallinity was calculated using Equation (1):(1)$$\%Crystallinity=(\frac{\sum Integrated\;peak\;areas\;of\;sample}{\sum Integrated\;peak\;areas\;of\;commercial\;Na-A\;zeolite})\times 100$$

Commercial Na-A zeolite was purchased from FUJIFILM Wako Pure Chemical Corporation (Osaka, Japan) and used as a standard in crystallinity calculations. Based on the XRD results, when the NaOH/kaolin mass ratio mounted from 0.6 to 0.9, the crystallinity of the samples increased from approximately 64% (ZK06) to 84% (ZK09).

### 2.2. Adsorption Performance and Characterization

For batch adsorption experiments, stock solutions were prepared using strontium nitrate (Sr(NO_3_)_2_, EP grade, Samchun, Korea), and deionized water was used in all experiments. A total of 0.05 g of ZK06 and ZK09 were added to conical centrifuge tubes (Falcon, 352070) containing 50 mL of the Sr^2+^ stock solution at 25 °C. At the same time, the pH was adjusted to 5.0 by the addition of 0.1 M HCl or NaOH. The prepared samples were stirred at 25 °C and 180 rpm in an orbital shaker (VS-8480SF, Vision Scientific Co., Ltd., Daejeon, Korea). After stirring and reaction, the samples were centrifuged at 3000× *g* rpm for 3 min using a centrifuge (VS-5500i, Vision Scientific Co., Ltd., Daejeon, Korea). The supernatants were filtered, and Sr^2+^ ion concentrations were determined using ICP-OES (Optima 8300, Perkin Elmer, Waltham, MA, USA).

#### 2.2.1. Adsorption Kinetic Models

To determine the kinetic characteristics of Sr^2+^ on the samples in the aqueous solution and to fit the experimental data, the pseudo-first- and pseudo-second-order kinetic models were employed. The pseudo-first-order model is expressed as follows [18]:(2)$$\frac{dq}{dt}=k_{1}(q_{e}-q),$$
where *q* is the adsorption capacity (mmol/g) at the time, *q_e_* is the adsorption capacity (mmol/g) after equilibrium, and *k*_1_ is the rate constant of adsorption (1/min). With the boundary conditions at *t =* 0 to *t = t* and *q =* 0 to *q* = *q*, Equation (2) can be integrated and yields
(3)$$\ln(q_{e}-q_{t})=\ln(q_{e})-k_{1}t.$$

The pseudo-second-order model for the adsorption can be expressed as follows [13]:(4)$$\frac{dq}{dt}=k_{2}(q_{e}-q{)}^{2}.$$

With the boundary conditions at *t =* 0 to *t = t* and *q* = 0 to *q* = *q*, Equation (4) can be integrated and yields
(5)$$\frac{t}{q_{t}}=\frac{1}{k_{2}q_{e}^{2}}+\frac{1}{q_{e}}t,$$
where *k*_2_ is the adsorption rate constant (g/mg·min) determined from Equation (5), and a linear equation is obtained.

#### 2.2.2. Adsorption Isotherm Models

To determine the adsorption equilibrium data for Sr^2+^ on the samples in aqueous solutions and to fit the experimental data, the Langmuir and Freundlich isotherm models were employed. The Langmuir model can be expressed as follows [19]:(6)$$q_{e}=\frac{k_{L}q_{m}C_{e}}{1+k_{L}C_{e}},$$
where *k_L_* is a Langmuir constant (L/mg) and *q_m_* is the maximum adsorption capacity (mmol/g).

The Freundlich model can be expressed as follows [20]:(7)$$q_{e}=k_{F}{C_{e}}^{\frac{1}{n}}$$
where *k_F_* is the Freundlich constant (mmol/g) (L/mmol)^1/*n*^ and 1/*n* is a Freundlich constant related to the adsorption strength.

## 3. Results

### 3.1. Adsorption Kinetics of Sr^2+^

The experimental adsorption data for Sr^2+^ using ZK06 and ZK09 analyzed by model fitting with the pseudo-first- and second-order kinetic models are displayed in Figure 3. The plots show that the adsorption capacities of Sr^2+^ onto the samples increased rapidly within the initial 15 min, and the adsorption reached equilibrium after 120 min. The adsorption kinetic parameters for Sr^2+^ onto the samples obtained from the experimental data obtained using Equations (2) and (4) are listed in Table 1.

From the pseudo-first-order kinetic plots, ln(*q_e_* − *q*) against *t* yields linear lines with slopes of *k*_1_ and intercepts of ln(*q_e_*), whereas from the second-order kinetic plots, *t*/*q_t_* against *t* produces linear lines with slopes of 1/*q_e_* and intercepts of 1/*k*_2_*q_e_*^2^. Furthermore, *q_e_*, *k*_1_, and *k*_2_ values are obtained from the slopes and intercepts of these kinetic models. Compared to the equilibrium adsorption capacity (*q_e,exp_*) obtained directly from experiments, as summarized in Table 1, the *q_e_* values estimated from the kinetic models were similar to the *q_e,exp_* values obtained from the pseudo-second-order model but were significantly different from those obtained from the pseudo-first-order model. In addition, the coefficient of determination, r^2^, was calculated as 0.9849 and 0.9965 with the pseudo-second-order model, and the results were relatively higher than those of the pseudo-first-order model. Therefore, the pseudo-second-order model can be considered more suitable in terms of characterizing the adsorption kinetics of Sr^2+^.

### 3.2. Adsorption Isotherms of Sr^2+^

The results of the adsorption isotherm experiment and the corresponding model fittings for Sr^2+^ are shown in Figure 4. The experimental adsorption isotherm data were analyzed using the Langmuir and Freundlich models, and the calculated parameters are listed in Table 2. Non-linear and linear isotherm models were compared based on parameters with coefficient of determination (r^2^). The r^2^ values of the Langmuir model were computed to be 0.9980 and 0.9963, which were comparatively higher than those of the Freundlich model. Accordingly, the Langmuir model is evaluated to be more suitable for the characterization of isotherms for Sr^2+^.

The values of the maximum adsorption capacities, *q_m_*, of ZK06 and ZK09 for Sr^2+^ drawn from the Langmuir model were 1.90 and 2.42 mmol/g, respectively. Moreover, when natural zeolite was used, the *q_m_* for Sr^2+^ obtained using the Langmuir model was 0.11 mmol/g [14]. Using the fusion method from CFA, Murukutti and Jena [21] calculated the *q_m_* of a synthesized Na-A zeolite for Sr^2+^ to be 1.09 mmol/g. The *q_m_* values of ZK06 and ZK09 for Sr^2+^ were approximately 73.6% and 93.4%, respectively, compared to 2.59 mmol/g of commercial Na-A zeolite [2]. Therefore, the Na-A zeolites synthesized from kaolin used in this study can be evaluated as efficient adsorbents for the removal of Sr^2+^.

## 4. Discussion

The high adsorption capacity of zeolites results from their low density, high porosity, high crystallinity (structure stability), and uniform-sized channels [22,23]. Choi and Lee [24] estimated the maximum crystallinity of a Na-A zeolite hydrothermally synthesized using CFA to be less than 60%, and the results of Sr^2+^ adsorption experiments confirmed that the optimal *q_m_* value was 1.51 mmol/g. Moreover, comparing the *q_m_* for Sr^2+^ of Na-A zeolites synthesized with high crystallinity under various conditions of synthesis [25,26,27,28] and the values of 1.90 mmol/g for ZK06 and 2.42 mmol/g for ZK09 obtained in this study, it was found that they had higher adsorption capacities compared to those reported in previous studies. Therefore, limitations in terms of crystallinity and ion adsorption capacity can be resolved by synthesizing Na-A zeolites using eco-friendly and low-cost natural minerals.

Under conditions where Sr and Ca ions coexist, the adsorption capacity of Sr obtained using a commercial Na-A zeolite tended to decrease slightly when compared to that of the capacity obtained in an aqueous solution with single Sr^2+^. However, even when Na^+^, K^+^, Ca^2+^, and Mg^2+^ were present in the solution, the selectivity for Sr^2+^ by the adsorption of Na-A zeolite was higher than that of other ions [29,30].

## 5. Conclusions

Na-A zeolites synthesized from kaolin were used as adsorbents for removing radioactive strontium from aqueous solutions. The experimental adsorption data for Sr^2+^ exhibited *q_m_* values of 1.90 and 2.42 mmol/g for the synthesized zeolites having NaOH/kaolin mass ratios of 0.6 and 0.9, respectively, and the Langmuir isotherm model suitably described the adsorption isotherm characteristics of the adsorbents. Based on these *q_m_* values, ZK06 and ZK09 were evaluated to have adsorption capacities of approximately 73.6% and 93.4%, respectively, compared with the commercial Na-A zeolite. Overall, synthesized zeolites from natural inorganic materials using the fusion/hydrothermal method can be used as eco-friendly adsorbents to remove radioactive nuclides.

## Figures and Tables

**Figure 1 materials-17-00575-f001:**
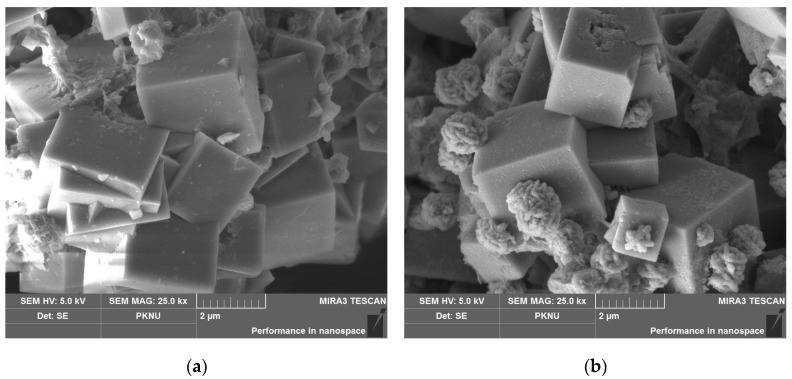
FE-SEM images of samples: (**a**) ZK06 with a median particle size of 1.79 μm and (**b**) ZK09 with a median particle size of 1.25 μm.

**Figure 2 materials-17-00575-f002:**
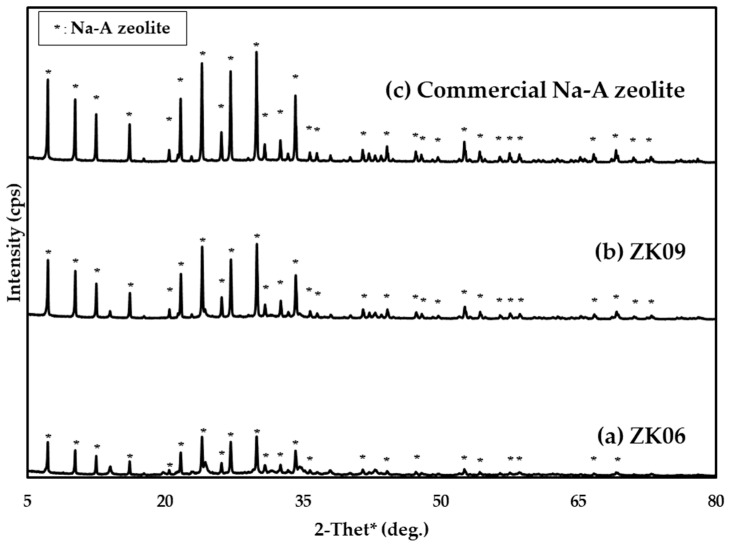
XRD patterns of hydrothermally synthesized Na-A zeolites with NaOH/kaolin mass ratios of (**a**) 0.6 (ZK06) and (**b**) 0.9 (ZK09). For calculating crystallinity, (**c**) commercial Na-A zeolite was used as a standard (reused with permission and modified after [15]).

**Figure 3 materials-17-00575-f003:**
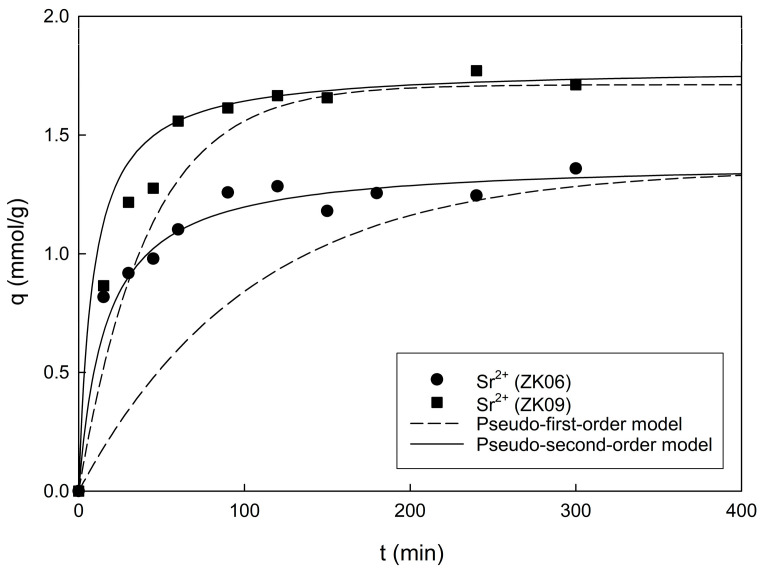
Experimental adsorption results fitted according to the kinetic models for Sr^2+^ using ZK06 and ZK09 with NaOH/kaolin mass ratios of 0.6 and 0.9 (dosage of adsorbents = 1 g/L).

**Figure 4 materials-17-00575-f004:**
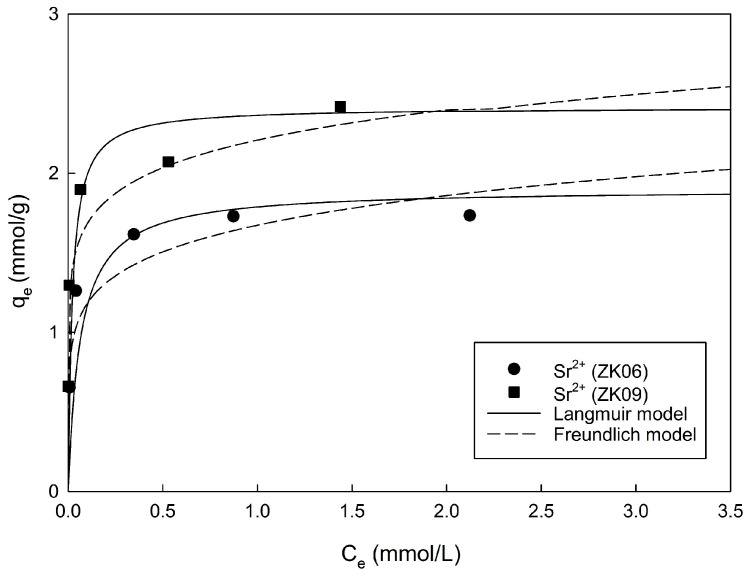
Experimental adsorption results fitted according to the isotherm models for Sr^2+^ using ZK06 and ZK09 with NaOH/kaolin mass ratios of 0.6 and 0.9 (dosage of adsorbents = 1 g/L).

**Table 1 materials-17-00575-t001:** Calculated kinetic parameters for the adsorption of Sr^2+^ onto ZK06 and ZK09 with NaOH/kaolin mass ratios of 0.6 and 0.9.

Sample	* *C*_0_	*q_e,exp_*	Pseudo-First-Order Model			Pseudo-Second-Order Model		
	(mmol/L)	(mmol/g)	*q_e_* (mmol/g)	*k*_1_ (1/h)	r^2^	*q_e_* (mmol/g)	*k*_2_ (mmol/g‧h)	r^2^
ZK06	2.56	1.36	0.5587	0.0334	0.6319	1.3918	0.0442	0.9849
ZK09	2.56	1.71	1.1294	0.0089	0.9157	1.7845	0.0648	0.9965

* *C*_0_: ion concentration at *t* = 0.

**Table 2 materials-17-00575-t002:** Calculated isotherm parameters for the adsorption of Sr^2+^ onto ZK06 and ZK09 with each NaOH/kaolin mass ratio of 0.6 and 0.9.

Sample	Langmuir Model			Freundlich Model		
	*q_m_* (mmol/g)	*k_L_* (L/mmol)	r^2^	*k_F_* (mmol/g∙(L/mmol)1/n)	1/n	r^2^
ZK06	1.90	0.8190	0.9980	1.6747	0.1521	0.8552
ZK09	2.42	0.1237	0.9963	2.2096	0.1185	0.9222

## Data Availability

Data are contained within the article.

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
