# Peer review of "The Removal of Strontium Ions from an Aqueous Solution Using Na-A Zeolites Synthesized from Kaolin"

_materials, 2024, doi:10.3390/ma17030575_

Round 1

Reviewer 1 Report

Comments and Suggestions for Authors

Materials 2807206

The removal Sr2+ in aqueous solution With Na-A zeolites, which were synthesized from kaolin and with different ratios 0.6 (ZH06) and 0.9 (ZH09). The absorption was achieved in120 min. The experimental absorption was estimated: pseudo-2nd-order kinetic and Langmuir isotherm models.

The introduction is complete and very interesting. The references are complete. The paper is well written and well organized but is necessary que the authors written with major detail which is the novelty of this manuscript. What is the new part and that was described previously in the literature, in particular compared with reference [11].

1)      The samples KH06 and KH09 has been previously described in the literature. Fig. 1 is of the authors, or previously presented in the literature? Ref. [11]

2)      In Fig. 2, the three compounds present the same XRD, the only difference is the intensity of peaks. Comment with major detail.

3)      Revise Fig. 4, add greater number of points.

4)      The Part 4, Discussion is repetitive. Add the part important in Conclusions.

Comments on the Quality of English Language

Moderate English is necessary

Author Response

[Reviewer 1]

  • Comments and Suggestions for Authors

The removal Sr2+ in aqueous solution With Na-A zeolites, which were synthesized from kaolin and with different ratios 0.6 (ZH06) and 0.9 (ZH09). The absorption was achieved in120 min. The experimental absorption was estimated: pseudo-2nd-order kinetic and Langmuir isotherm models.

The introduction is complete and very interesting. The references are complete. The paper is well written and well organized but is necessary que the authors written with major detail which is the novelty of this manuscript. What is the new part and that was described previously in the literature, in particular compared with reference [11].

: Thank you for your time and comments. Modified parts in the manuscript are marked in red.

1) The samples KH06 and KH09 has been previously described in the literature. Fig. 1 is of the authors, or previously presented in the literature? Ref. [11]

: We used non-identical SEM images of samples from previous papers (before revision [11] and after revision [15]).

2) In Fig. 2, the three compounds present the same XRD, the only difference is the intensity of peaks. Comment with major detail.

: We added the following comment to lines 81-87:

“To determine that the crystal structures of ZK06 and ZK09 were formed through the synthesis process, their X-ray diffraction (XRD) patterns were compared with the commercial Na-A zeolite as the standard. Figure 2 shows that as the NaOH/kaolin mass ratio increases, its XRD pattern and peak intensity become closer to that of the commercial Na-A zeolite (H74Al11.8Na13.1O85Si12.2). When the XRD pattern of ZK06 has a low NaOH/kaolin mass ratio, low-intensity peaks remain, suggesting the presence of amorphous components that have not yet been crystallized.”

3) Revise Fig. 4, add greater number of points.

: To clearly show the line fitting of the adsorption results, the results above 3.5 mmol/L were cropped. Please understand.

4) The Part 4, Discussion is repetitive. Add the part important in Conclusions.

: We revised Sections 4 and 5, excluding duplicates. Please check lines 191-217.

  • Comments on the Quality of English Language

Moderate English is necessary.

: To ensure a smooth flow of text, the English text was re-edited by ‘Editage’. A certificate of English proofreading has been submitted as an attachment.

Reviewer 2 Report

Comments and Suggestions for Authors

The authors have presented an interesting study into the synthesis of zeolites from kaolin. this is interesting for various applications and the cost-effectivness of this approach for diverse engineering applications merits further scrutiny. 

There are several weaknesses identified here:

The cost-effectiveness of the zeolite synthesis is insufficiently presented and should be examined with reference to commercial sources and other methods. 

The crystallinity as calculated by equation 1, is not well-supported. If the peak area was greater than the reference sample, the crystallinity would be greater than 100%, which doesn't make sense.  A better approach towards assessing the efficacy of the zeolite crystallisation is needed. 

The adsorption of Sr is interesting. Why is this chosen as the main application for this aluminosilicate? Is there a reason to expect that this material will perform particularly well in this application? 

Sr2+ ions are very similar to Ca2+ ions. I would expect that the adsorption of calcium ions would saturate adsorption sites. What is the expected ion speciation in the treated water?

Scale dependent adsorption would also be interesting. The surface area of the zeolite is scale dependent. This should be discussed. Ca ions and other ions in the aqueous solution may have a smaller adsorption cross section than strontium. This merits some discussion. 

Comments on the Quality of English Language

The English should be revised to provide some more convincing descriptions of the comparative synthesis and performance of the material.

Minor improvements to syntax and grammar would be beneficial. 

Author Response

[Reviewer 2]

  • Comments and Suggestions for Authors

The authors have presented an interesting study into the synthesis of zeolites from kaolin. this is interesting for various applications and the cost-effectivness of this approach for diverse engineering applications merits further scrutiny. There are several weaknesses identified here:

: Thank you for your time and comments. Modified parts in the manuscript are marked in red.

The cost-effectiveness of the zeolite synthesis is insufficiently presented and should be examined with reference to commercial sources and other methods.

: We added the following comment to lines 47-53:

“Among the various synthesis methods for producing Na-A zeolites, hydrothermal and fusion/hydrothermal methods are widely used in the synthesis of inexpensive raw materials such as coal fly ash, blast-furnace slag, waste sludge, lithium slag, and natural minerals [10]. Kaolin is a low-cost raw material and is abundant across the world. When kaolin is used as a synthetic raw material, Na-A zeolites can be obtained by dis-solving metakaolin in a sodium hydroxide (NaOH) solution without adding any other silica or alumina sources [11].”

The crystallinity as calculated by equation 1, is not well-supported. If the peak area was greater than the reference sample, the crystallinity would be greater than 100%, which doesn't make sense.  A better approach towards assessing the efficacy of the zeolite crystallisation is needed.

: The XRD 2 theta range, which was the basis for crystallinity calculation, has been added in detail to lines 88-92 as follows.

“The crystallinity for evaluating the synthesis efficiency of the sample was approximated by taking the ratio of the sum of the integrated peak areas of the sample to the sum of the integrated peak areas of commercial Na-A zeolite that appear intrinsically at 2θ values of 7.20°, 10.18°, 12.46°, 16.10°, 21.66°, 23.98°, 26.10°, 27.10°, 29.92°, 30.82°, 32.52°, and 34.16° [16, 17].”

The adsorption of Sr is interesting. Why is this chosen as the main application for this aluminosilicate? Is there a reason to expect that this material will perform particularly well in this application?

: An explanation for this comment has been added to lines 37-45.

“Na-A zeolite is one of the low-silica zeolites with an Si/Al content ratio of 1 that has a hexahedron structure [8]. While high-alumina zeolites are unsuitable as catalysts due to their lack of acidity, Na-A zeolites are appropriate for adsorption reactions due to the large density of cation-capturing sites on the framework. Owing to their characteristically high cation-exchange capacity (CEC), Na-A zeolites are widely used for removing toxic metal ions and radionuclides from aqueous solutions in wastewater treatment. In particular, since the pore apertures of Na-A zeolites are in the range of 0.23 nm to 0.42 nm, they can be considered to have a framework structure suitable for capturing Sr2+, which has a hydration radius of approximately 0.41 nm [9].”

Sr2+ ions are very similar to Ca2+ ions. I would expect that the adsorption of calcium ions would saturate adsorption sites. What is the expected ion speciation in the treated water?

Scale dependent adsorption would also be interesting. The surface area of the zeolite is scale dependent. This should be discussed. Ca ions and other ions in the aqueous solution may have a smaller adsorption cross section than strontium. This merits some discussion.

: Thank you for the interesting comment. We aim to follow up on the comments and have added explanations in Section 4, lines 203-207.

“Under conditions where Sr and Ca ions coexist, the adsorption capacity of Sr obtained using a commercial Na-A zeolite tended to decrease by 31.8% to 1.2 mmol/g, when compared to 1.76 mmol/g obtained in an aqueous solution with single Sr2+. However, even when Na+, K+, Ca2+, and Mg2+ were present in the solution, the selectivity for Sr2+ by the adsorption of Na-A zeolite was higher than that of other ions [28].”

  • Comments on the Quality of English Language

The English should be revised to provide some more convincing descriptions of the comparative synthesis and performance of the material.

Minor improvements to syntax and grammar would be beneficial.

: To ensure a smooth flow of text, the English text was re-edited by ‘Editage’. A certificate of English proofreading has been submitted as an attachment.

Reviewer 3 Report

Comments and Suggestions for Authors

Comments on trapping of strontium ions by Kaolin-derived zeolites

The authors present data characterizing the sequestering of strontium ions by zeolites derived from kaolin by reaction with NaOH. Two forms were studied, prepared with 0.6 and 0.9 mass ratios (called ZK06 and ZK09). Crystallinity is defined relative to commercial Na-A zeolite by the relative values of integrated peak areas in the X ray diffraction spectrum over the range of 2 x theta from 7 to 34 degrees. ZK06 is assigned 64% and ZK09 84% of the crystallinity of commercial Na-A zeolite. Kinetic fit to adsorption data is better for the second-order model, which matches adsorption capacity within 3% (R2 > 0.98). The Langmuir isotherm fits data better than the Freundlich isotherm, and provides an estimate of maximum adsorption capacity (mmol Sr/gram zeolite) of 1.90 for ZK06 and 2.42 for ZK09, to be compared with the value of 2.59 for commercial Na-A zeolite. The parallel between degree of crystallinity and effectiveness of removal of strontium seems well established.

This is sound work and establishes that the kaolin-derived zeolites can be effective in removal of hazardous strontium from aqueous solution. This adds to our knowledge of decontamination processes. 

It is not clear to me why Na-A zeolite should not be used for this purpose, or whether the materials prepared by the authors has some advantage. A question that may be outside the scope of this investigation is whether treatment of strontium-contaminated water with the sequestering agent would be cost-effective in practical settings.

Author Response

[Reviewer 3]

  • Comments and Suggestions for Authors

The authors present data characterizing the sequestering of strontium ions by zeolites derived from kaolin by reaction with NaOH. Two forms were studied, prepared with 0.6 and 0.9 mass ratios (called ZK06 and ZK09). Crystallinity is defined relative to commercial Na-A zeolite by the relative values of integrated peak areas in the X ray diffraction spectrum over the range of 2 x theta from 7 to 34 degrees. ZK06 is assigned 64% and ZK09 84% of the crystallinity of commercial Na-A zeolite. Kinetic fit to adsorption data is better for the second-order model, which matches adsorption capacity within 3% (R2 > 0.98). The Langmuir isotherm fits data better than the Freundlich isotherm, and provides an estimate of maximum adsorption capacity (mmol Sr/gram zeolite) of 1.90 for ZK06 and 2.42 for ZK09, to be compared with the value of 2.59 for commercial Na-A zeolite. The parallel between degree of crystallinity and effectiveness of removal of strontium seems well established.

This is sound work and establishes that the kaolin-derived zeolites can be effective in removal of hazardous strontium from aqueous solution. This adds to our knowledge of decontamination processes.

: Thank you for your kind and knowledgeable comments. The text has been modified based on your suggestions and marked in red.

It is not clear to me why Na-A zeolite should not be used for this purpose, or whether the materials prepared by the authors has some advantage. A question that may be outside the scope of this investigation is whether treatment of strontium-contaminated water with the sequestering agent would be cost-effective in practical settings.

: We added the following comment to lines 47-53:

“Among the various synthesis methods for producing Na-A zeolites, hydrothermal and fusion/hydrothermal methods are widely used in the synthesis of inexpensive raw materials such as coal fly ash, blast-furnace slag, waste sludge, lithium slag, and natural minerals [10]. Kaolin is a low-cost raw material and is abundant across the world. When kaolin is used as a synthetic raw material, Na-A zeolites can be obtained by dis-solving metakaolin in a sodium hydroxide (NaOH) solution without adding any other silica or alumina sources [11].”

Round 2

Reviewer 2 Report

Comments and Suggestions for Authors

The manuscript has been improved following reviewer input